Squeezing water from a stone: high-throughput sequencing from a 145-year old holotype resolves (barely) a cryptic species problem in flying lizards

McGuire Jimmy A. mcguirej@berkeley.edu 1 2
Cotoras Darko D. 3 4 13
O’Connell Brendan 5 14
Lawalata Shobi Z.S. 1 2 6
Wang-Claypool Cynthia Y. 1 2
Stubbs Alexander 1 2
Huang Xiaoting 7
Wogan Guinevere O.U. 8
Hykin Sarah M. 1 2
Reilly Sean B. 1 2
Bi Ke 9
Riyanto Awal 10
Arida Evy 10
Smith Lydia L. 1 2
Milne Heather 5
Streicher Jeffrey W. 11
Iskandar Djoko T. 12
1 Museum of Vertebrate Zoology, University of California , Berkeley , CA , United States of America
2 Department of Integrative Biology, University of California , Berkeley , CA , United States of America
3 Department of Ecology and Evolutionary Biology, University of California , Santa Cruz , CA , United States of America
4 Entomology Department, California Academy of Sciences , San Francisco , CA , United States of America
5 Department of Biomolecular Engineering and Bioinformatics, Baskin School of Engineering, University of California , Santa Cruz , CA , United States of America
6 Jalan Hayam Wuruk, United in Diversity Foundation , Jakarta , Indonesia
7 Key Laboratory of Marine Genetics and Breeding, Ocean University of China , Qingdao , China
8 Department of Environmental Science, Policy, and Management, University of California , Berkeley , CA , United States of America
9 Computational Genomics Resource Laboratory, California Institute for Quantitative Biosciences, University of California , Berkeley , CA , United States of America
10 Museum Zoologicum Bogoriense, Research Center for Biology-The Indonesian Institute of Sciences , Cibinong , Indonesia
11 Department of Life Sciences, The Natural History Museum , London , United Kingdom
12 School of Life Sciences and Technology, Institute of Technology Bandung , Bandung , West Java , Indonesia
13 Current affiliation:  Center for Comparative Genomics, California Academy of Sciences , San Francisco , CA , United States of America
14 Current affiliation:  Department of Medical & Molecular Genetics, School of Medicine, Oregon Health & Science University , Portland , OR , United States of America
Edwards Scott
Electronic publication date: 2018 Mar 20
Publication date: 2018
Volume: 6
Electronic Location ID: e4470
Received 2017 Oct 26; Accepted 2018 Feb 16
Copyright: ©2018 McGuire et al.
Copyright year: 2018
Copyright holder: McGuire et al.
License: This is an open access article distributed under the terms of the Creative Commons Attribution License, which permits unrestricted use, distribution, reproduction and adaptation in any medium and for any purpose provided that it is properly attributed. For attribution, the original author(s), title, publication source (PeerJ) and either DOI or URL of the article must be cited.
License URL: https://creativecommons.org/licenses/by/4.0/

Keywords: Draco, Formalin-fixation, Ancient DNA, Phylogeny, Taxonomy

Funding: National Geographic Society and the National Science Foundation DEB 1652988 DEB 1258185 This work was supported by the National Geographic Society and the National Science Foundation (DEB 1652988 and DEB 1258185). The funders had no role in study design, data collection and analysis, decision to publish, or preparation of the manuscript.

==============================
We used Massively Parallel High-Throughput Sequencing to obtain genetic data from a 145-year old holotype specimen of the flying lizard, Draco cristatellus. Obtaining genetic data from this holotype was necessary to resolve an otherwise intractable taxonomic problem involving the status of this species relative to closely related sympatric Draco species that cannot otherwise be distinguished from one another on the basis of museum specimens. Initial analyses suggested that the DNA present in the holotype sample was so degraded as to be unusable for sequencing. However, we used a specialized extraction procedure developed for highly degraded ancient DNA samples and MiSeq shotgun sequencing to obtain just enough low-coverage mitochondrial DNA (721 base pairs) to conclusively resolve the species status of the holotype as well as a second known specimen of this species. The holotype was prepared before the advent of formalin-fixation and therefore was most likely originally fixed with ethanol and never exposed to formalin. Whereas conventional wisdom suggests that formalin-fixed samples should be the most challenging for DNA sequencing, we propose that evaporation during long-term alcohol storage and consequent water-exposure may subject older ethanol-fixed museum specimens to hydrolytic damage. If so, this may pose an even greater challenge for sequencing efforts involving historical samples.

Introduction

The advent of Massively Parallel High-Throughput Sequencing (HTS) has dramatically altered the manner in which geneticists conduct their research. This is certainly true for molecular phylogeneticists and population geneticists, who now routinely have access to large multilocus genetic datasets for non-model organisms. Because HTS using the Illumina platform involves sequencing of small fragments of DNA, this approach offers the potential to access previously unattainable genome-scale sequence data even for degraded historical samples (e.g., Prüfer et al., 2014; Palkopoulou et al., 2015). Millions of fluid-preserved specimens in museum collections predate the development of allozyme and DNA sequencing technologies, and thus lack specially preserved tissue samples for genetic analysis. Formalin-fixed fluid specimens, usually having highly fragmented and cross-linked DNA, are often refractory to sequencing efforts using traditional Sanger sequencing. However, recent studies have shown that it is possible to obtain genomic DNA sequences from some of these fluid-preserved museum specimens. Hykin, Bi & McGuire (2015) demonstrated that low-coverage genomic sequences could be recovered from a 30-year old formalin-fixed museum specimen, though they were unsuccessful with a ∼100-year old specimen. Ruane & Austin (2016) sequenced Ultra-Conserved Elements (UCEs) from both formalin-fixed (n = 11) and ethanol-fixed (n = 10) museum specimens, including one sample that was collected between 1878 and 1911. Both had mixed success, with the quantity of DNA recovered in the extraction stage likely playing the largest role in the performance of their sequencing efforts. Notably, the samples that failed in the Ruane & Austin (2016) experiment included subsets of both their formalin- (seven of 16) and alcohol-fixed (four of five) samples, indicating that old alcohol-preserved museum specimens are not necessarily less problematic than those initially fixed with formalin. This is surprising given that contemporary tissue samples earmarked for genetic analysis are routinely stored in 95% ethanol.

Our objective in this study was to address an otherwise intractable problem in flying lizard taxonomy using Illumina HTS and ancient DNA methods for a 145-year old fluid-preserved holotype specimen. The nettlesome taxonomic issue involves a small clade of poorly known flying lizards (genus Draco, Agamidae) that, for reasons outlined below, was unlikely to be resolved without obtaining genetic data from the holotype specimen of one of the constituent species, Draco cristatellus. Determining species limits within this small clade (the Draco fimbriatus group) has proven challenging for taxonomists, and we first describe the convoluted taxonomic history of the clade as a justification for our ultimate solution to this question involving HTS. The D. fimbriatus group currently includes four recognized species: D. abbreviatus, D. cristatellus, D. fimbriatus, and D. maculatus. This taxonomic framework is based on Manthey (2008) and was followed by the widely utilized Reptile Database (Uetz, 2006). For reasons that we will describe in a subsequent paper, we instead utilize an alternative taxonomy that includes D. cristatellus, as well as D. fimbriatus (=D. abbreviatus above), D. hennigi, D. punctatus (=D. fimbriatus above), and D. maculatus. We note that our taxonomy differs from that of Manthey (2008) primarily as a consequence of having information that indicates that the type locality of D. fimbriatus is the Malay Peninsula rather than Java. We further note that our recognition of both D. cristatellus and D. punctatus is tentative, as a primary objective of this paper is to resolve whether these are in fact distinct species.

The Draco fimbriatus group is taxonomically challenging. Although Draco maculatus is abundant, easily sampled in the field, and easily distinguished from other members of the group based on external phenotype, the remaining members of this clade are only rarely encountered, with relatively few specimens represented in museum collections. These species are canopy specialists (McGuire, 2003), making them more difficult to detect and more challenging to collect than other Draco taxa. Furthermore, the species comprising the D. fimbriatus group, as well as several other Draco clades are primarily distinguished on the basis of differences in coloration of their display structures (dewlap and patagia for most Draco taxa, just the dewlap among the relevant members of the D. fimbriatus group). For example, two major clades—the ‘Philippines volans group’ (McGuire & Alcala, 2000) and the ‘Draco lineatus group’ (McGuire et al., 2007)—are each composed of multiple species that are primarily distinguished on the basis of coloration. Because coloration fades in preservative, recognizing species-specific coloration characteristics generally requires experience with the species in the field and/or access to color imagery of the specimen in life. Thus, as museum specimens, the members of these clades can become functionally cryptic sympatric species. In summary, for the D. fimbriatus group, the paucity of museum specimens, and the rarity with which specimens are observed in the field from throughout their collective ranges by single observers, has greatly impeded taxonomic progress.

Within the Draco fimbriatus group, a particularly challenging issue relates to the taxonomic standings of D. cristatellus Günther, 1872 and D. punctatus Boulenger, 1900. Draco cristatellus was described based on a single specimen collected by Mr. Alfred Hart Everett in Matang, Sarawak between 1869 and March of 1872 (when Günther’s manuscript describing the species was submitted for publication). Although Everett collected the type specimen, the Trustees of the British Museum purchased it from Mr. W. Cutter, thereby making it available to Günther for description (see Günther, 1872). Because Günther presumably did not see the living specimen, he did not evaluate the coloration of the dewlap in life, which is essential for species identification within this group. Günther described the dewlap as ‘golden-yellow, with a brown anterior edge’, presumably from its preserved state. Subsequently, Boulenger (1900) described D. punctatus from Bukit Larut on the Malay Peninsula, noting that the dewlap was lemon yellow in coloration. Although he did not attempt to diagnose D. punctatus from D. cristatellus, Boulenger (1900) was clearly aware of the latter species and explicitly considered his D. punctatus holotype to be taxonomically distinct. Indeed, Boulenger (1900) noted that he had examined a second specimen of D. punctatus from Sarawak that was also collected by Everett, remarking that he had previously referred that second specimen to D. cristatellus. Boulenger (1900) might be the last author to have had a clear idea about the taxonomic distinctiveness of D. cristatellus and D. punctatus, and it is a pity that he did not identify the character differences that he used to render his taxonomic decision. Although De Rooij (1915) recognized D. fimbriatus, D. cristatellus and D. punctatus as distinct species, subsequent authors synonymized one or more members of the group. Hennig (1936) synonymized D. cristatellus with D. fimbriatus, while continuing to recognize D. punctatus. In his monographic Draco taxonomic study, Musters (1983) opted to synonymize both D. cristatellus and D. punctatus with D. fimbriatus. In his competing taxonomic treatment, Inger (1983) recognized two species, D. cristatellus and D. fimbriatus, as valid species, but placed D. punctatus in the synonymy of D. cristatellus. Inger’s (1983) recognition of two species was based in part on Grandison’s (1972) report on two sympatric D. fimbriatus group species with distinct dewlap colorations on Gunung (Mt.) Benom on the Malay Peninsula. Whereas Grandison (1972) identified the two sympatric species as D. fimbriatus and D. punctatus (without commenting on the status of the Bornean D. cristatellus), Inger (1983) instead opted to treat D. punctatus as a synonym of D. cristatellus. This sensible decision was presumably made on the basis of the similar dewlap colorations of the D. cristatellus and D. punctatus holotypes (‘golden-yellow, with a brown anterior edge’ vs. ‘lemon yellow’). Inger (1983) furthermore attempted to differentiate his conceptions of D. fimbriatus and D. cristatellus using a statistical analysis of eight linear measurements and scale counts. Although he successfully sorted his sample into two groups on the basis of overlapping but significantly distinct character state differences, his a priori placement of D. punctatus in the synonymy of D. cristatellus effectively precluded the possibility that three species –D. cristatellus, D. fimbriatus, and D. punctatus—might all co-occur on the Greater Sunda Shelf (and particularly on Borneo). Here we address this open question taking advantage of two critical developments: (1) the acquisition and analysis of a key specimen (TNHC 56763) obtained by JAM from Santubong, Sarawak, Malaysian Borneo in 1996, and (2) an analysis of the 145-year old D. cristatellus holotype using ancient DNA extraction methods and HTS on the Illumina platform. We establish the following hypothesis-testing framework: In Hypothesis 1, D. cristatellus and D. punctatus are synonyms, together representing a single species distinct from D. fimbriatus and TNHC 56763. In Hypothesis 2, D. cristatellus and D. fimbriatus are synonyms. In Hypothesis 3, D. cristatellus is a species distinct from D. punctatus and D. fimbriatus but conspecific with TNHC 56763. Finally, in Hypothesis 4, D. cristatellus, D. fimbriatus, D. punctatus, and TNHC 56763 all represent distinct species, with TNHC 56763 representing a fourth sympatric species on Borneo. The only way to conclusively test these alternative hypotheses is to obtain informative genetic data from the holotype specimen of D. cristatellus for comparison with TNHC 56763, as well as representative specimens of D. fimbriatus and D. punctatus.

Materials and Methods

(a) DNA extraction and sequencing from the Draco cristatellus holotype

We obtained from the Natural History Museum in London liver tissue from the holotype of Draco cristatellus (specimen BMNH 1872.2.19.4). This specimen was originally collected and prepared prior to March 1872, well before the advent of formalin-fixation. At that time, the standard practice for fluid preservation of reptiles, amphibians, and fishes was direct preservation in “pure spirits of wine” (Günther, 1880). Thus, the holotype was most likely initially fixed in 90–100% ethanol (=Günther’s “pure spirits of wine”) and never exposed to formalin. Nevertheless, we opted to perform our initial DNA extraction using the methodology described in Hykin, Bi & McGuire (2015) for formalin-fixed tissues, with the goal being to perform an exome-capture experiment with this sample. The Hykin, Bi & McGuire (2015) procedure involves a series of initial ethanol washes followed by treatment in a heated alkali buffer solution to break cross-linkages before standard phenol-chloroform extraction. When this extraction returned a very low (potentially zero) yield, we performed a second pair of phenol-chloroform extractions involving phase-lock gel tubes followed by SPRI bead clean-up. This second round of extractions was performed with and without exposure to heated alkali solution. These extractions also failed to return sufficient DNA to move forward with library preparation. Despite minimal DNA yield, we made an attempt to PCR-amplify and sequence a short fragment of the mitochondrial ND2 gene from both DNA extracts. These experiments resulted in the amplification and sequencing of human ND2 in two separate experiments. Both of our low-yield DNA extractions were then sent to MYcroarray Inc. in Ann Arbor, MI where they were subjected to an extra silica purification designed for low-concentration fragment retention, and prepared as libraries. However, the library preparation retrieved only artifact and we did not proceed to targeted enrichment of selected exons or sequencing.

At this stage, we engaged with a lab specializing in genetic analysis of ancient DNA samples, with extraction and sequencing performed in a facility specifically designed for work with ancient samples (the Green/Shapiro Lab at UC Santa Cruz). No reptile work had previously been done in this facility and all work followed lab standards for working with historical samples (Fulton, 2011). The DNA extraction protocol was based on Dabney et al. (2013), Tin, Economo & Mikheyev (2014), and Cotoras et al. (2017). An initial subsample of 40 mg of tissue was subdivided into ∼1 mm pieces and suspended in lysis buffer. The composition of the 100 mL lysis buffer aliquot was: 5.3 mL 1 M Tris-HCl (pH 8.0), 5.3 mL 0.2 M EDTA, 10.6 mL 20% Sarkosyl, 1 mL 2-mercaptoethanol, and 77.8 mL distilled water. The tissue was digested with a total of 1 mL of lysis buffer with 1 mg/mL proteinase K, initially incubated overnight at 56 °C, and then raised to 72 °C for 1 hr. The 72 °C incubation step was undertaken in case the sample had been exposed to formalin in order to reverse any potential crosslinks. Silica-based purification followed the centrifugation-based protocol described by Dabney et al. (2013). Briefly, 0.5 mL of 3 M sodium acetate was added to the lysate and them transferred to a tube with 13 mL of binding buffer. The binding buffer is prepared in a 50 mL tube by first adding 23.88 g of guanidine hydrochloride and then adding water to bring the volume to 30 mL. A key element of this purification protocol is the high salt concentration of this binding buffer, which enhances recovery of short DNA fragments. After complete dissolution of the guanidine hydrochloride, 25 µl of Tween-20 and sufficient isopropanol to bring the total volume to 50 mL were added. The mixture of sample, binding buffer, and sodium acetate was transferred into a Zymo extension reservoir attached to a MiniElute spin column. The spin column was then centrifuged for 10 min at 1,000 rpm, after which the spin column was transferred to a 1.5 mL Eppendorf tube. We performed a dry spin for 1 min at 13,000 rpm, followed by two washes with 750 µl of PE buffer (1 min spin at 6,000 rpm). To ensure the entire PE buffer was removed, we did a dry spin for 1 min at maximum speed. We eluted the purified extract in two volumes of 25 µl of TET. Each sample was centrifuged for 30 s at 13,200 RPM after 3–5 min of incubation. Because the elution displayed pigmentation, 25 µl of the extract was purified on a column filled with cross-linked polyvinylpyrrolidone (PVPP) (Arbeli & Fuentes, 2007). We also produced an extraction control consisting of lysis buffer that was subjected to the same set of procedures.

For genomic sequencing, we prepared three barcoded Illumina sequencing libraries (two for the holotype sample and one for the control) using the Meyer & Kircher (2010) protocol, starting with 5 µl of the PVPP purified DNA extraction. The same volume was used for the extraction control. The libraries were sequenced on an Illumina MiSeq machine using 150-cycle v3 chemistry (2 × 75). Following sequencing, adaptors were removed from reads and sequences were merged using SeqPrep2 (https://github.com/jeizenga/SeqPrep2). Default parameters were used with the exception of the following: -q 20 -L 30 -A AGATCGGAAGAGCACACGTC -B AGATCGGAAGAGCGTCGTGT. FastQC (http://www.bioinformatics.babraham.ac.uk/projects/fastqc/) confirmed that the sequence quality was good, with the normal base quality drop in the final five bases.

(b) Sanger sequence data for Draco fimbriatus group specimens

Our team has generated a large number of complete ND2 sequences for Draco specimens, including for 65 exemplars representing the D. fimbriatus group. These sequences were available for comparison with ND2 sequence fragments obtained from the D. cristatellus holotype. PCR-amplification was undertaken using the primers Metf1 and ALAr2, with cycle sequencing involving these external primers plus the internal primers Metf5 and ND2r6 (see McGuire & Kiew, 2001 for details).

(c) Exome-capture and screening of mitochondrial DNA

For another project, we generated an exome-capture data set using the MyBaits in-solution capture system for a set of 350 samples spanning all of Draco. This sample set included 14 D. fimbriatus group samples. The target loci for the exome capture include 1,400 exons and flanking sequences derived from transcriptome sequences (jointly representing 709 loci), which were supplemented with an additional 540 lizard-specific UCE loci. Libraries enriched for our target loci were barcoded and sequenced on an Illumina Hi Seq 4000. Although our experiment was specifically designed to avoid capturing mitochondrial genes, mitochondrial sequences are so abundant in genomic DNA extractions that some mitochondrial molecules inevitably find their way into the off-target by-catch (non-target DNA sequences that are obtained during an exome-capture experiment). We took advantage of this imperfect filter to obtain mitochondrial sequences for comparison with the D. cristatellus holotype. For TNHC 56763, we used Geneious version 8.1.7 (Kearse et al., 2012) to obtain a mostly complete representation of mitochondrial coding genes by mapping our raw exome capture data (including off-target sequences) to the complete mitochondrial genome of Acanthasaura armata available on GenBank (AB266452.1). A preliminary assessment of the identity of the sequences was performed with a BLAST search after collapsing duplicate sequences with fastx_collapser (http://hannonlab.cshl.edu/fastx_toolkit/commandline.html#fastx_ collapser_usage). The result of the BLAST search was visualized with the program MEGAN (Huson et al., 2007). Processed reads were mapped with BWA mem (Li & Durbin, 2009) against the reference partial mitochondrial genome of TNHC 56763. Duplicates were removed with samtools rmdup (Li et al., 2009). A total of 17 unique reads mapped against the reference after duplicate removal. They represent a total of 777 bp of the 8,114 bp reference. Most of the mapped regions had 1 × coverage and portions of four contigs had 2 × coverage. The average length of the mapped reads was 53 bp. Finally, for each of the 14 D. fimbriatus group samples included in our exome-capture experiment, we used Geneious to map our raw sequencing reads to 10 mitochondrial contigs obtained for the D. cristatellus holotype. The raw sequence data is available on the SRA database.

(d) Analysis of DNA sequence variation

Our analysis of DNA sequence variation included alignment of homologous DNA sequences and a simple count of nucleotide base substitutions between the Draco cristatellus holotype, sample TNHC 56763 from Santubong, Sarawak, and our selection of D. fimbriatus and D. punctatus samples from the Malay Peninsula, Sumatra, the Mentawai Islands, Java, and Borneo. Specimens examined are listed in Table 1. We also performed a heuristic parsimony analysis to obtain a phylogram for the D. fimbriatus group and performed a non-parametric bootstrap analysis with 1,000 replicates to assess branch support. We did not perform a more rigorous maximum likelihood or Bayesian analysis because our primary objective was to assess uncorrected relative branch lengths. Phylogenetic analyses were performed in PAUP version 4 (Swofford, 2002).

Table 1 Numbers of base substitutions and sequence divergence values between the Draco cristatellus holotype and 14 exemplars representing the D. fimbriatus group.

Base pair differences between the Draco cristatellus holotype and each of 14 D. fimbriatus group samples for six mitochondrial genes. For ND2, the data used for comparisons were generated using standard Sanger sequencing. For all other genes, the data were derived from exome-capture off-target sequences. Mean sequence divergence values relative to the holotype are provided for each gene for each species. The final column indicates the total number of base changes and percentage divergences from the holotype, with the caveat that these summary values do not account for the unique evolutionary rates that typify each of these mitochondrial genes.

	COX III	COI	COI	ND4L	ND5	ND2	Total (%)	
	contig3a	contig12a	contig12b	contig13	contig15	Sanger		
cristatellus TNHC 56763	0/43 (0%)	1/79 (1.3%)	1/81 (1.2%)	0/79 (0%)	0/64 (0%)	3/183 (1.6%)	5/547 (0.9%)	
punctatus Borneo TNHC 56766		7/79 (8.9%)	7/69 (10.1%)			30/183 (16.4%)	44/331 (13.3%)	
punctatus Borneo TNHC 56764		6/79 (7.6%)	8/81 (9.9%)	10/69 (14.5%)		28/183 (15.3%)	52/412 (12.6%)	
punctatus Malay Pen. LSUHC 5617	5/43 (11.6%)			10/79 (12.7%)		32/183 (17.5%)	47/305 (15.4%)	
punctatus Mentawai MVZ 270632	4/43 (9.3%)	10/79 (12.7%)	7/81 (8.6%)	12/79 (15.2%)		37/183 (20.2%)	70/465 (15.1%)	
punctatus Batu Ids MVZ 270636	4/43 (9.3%)	10/79 (12.7%)	7/81 (8.6%)	12/79 (15.2%)		35/183 (19.1%)	68/465 (14.6%)	
punctatus Sumatra MVZ 270835	4/43 (9.3%)	9/79 (11.4%)	7/81 (8.6%)	11/79 (13.9%)		35/183 (19.1%)	66/465 (14.2%)	
punctatus Banyak Ids MVZ 270829	4/43 (9.3%)	10/79 (12.7%)	7/81 (8.6%)	9/59 (15.3%)		37/183 (20.2%)	67/465 (14.4%)	
fimbriatus Mal Pen TNHC 57954	4/43 (9.3%)	7/71 (9.9%)	7/81 (8.6%)	9/79 (11.4%)	12/64 (18.8%)	29/183 (15.8%)	68/521 (13.1%)	
fimbriatus Mal Pen TNHC 58565	4/43 (9.3%)	10/79 (12.7%)	7/80 (8.8%)	9/79 (11.4%)	12/64 (18.8%)		42/345 (12.2%)	
fimbriatus Sumatra MZB Lace.14276	9/79 (11.4%)	5/56 (8.9%)	8/79 (10.1%)		32/183 (17.5%)	54/397 (13.6%)	
fimbriatus Sumatra MVZ 239473		9/79 (11.4%)	8/81 (9.9%)	8/79 (10.1%)	11/64 (17.2%)		36/303 (11.9%)	
hennigi LSUMZ 81446	11/79 (13.9%)		8/79 (10.1%)	12/64 (18.8%)	23/183 (12.6%)	54/405 (13.3%)	
hennigi LSUMZ 81447	11/79 (13.9%)		8/79 (10.1%)	13/64 (20.3%)	23/183 (12.6%)	55/405 (13.6%)	
						cristatellus	5/547 (0.9%)	
						punctatus	414/2908 (14.2%)	
						fimbriatus	200/1566 (12.8%)	
						hennigi	90/700 (12.9%)	

(e) Data availability and Permits

Sanger sequence data are available at GenBank (SRP135680) and a matrix for the mitochondrial ND2 gene for the D. fimbriatus group is included as supplemental materials. This research was undertaken in accordance with UC Berkeley Animal Use Protocol Number AUP-2014-12-6954. Fieldwork was undertaken with research permits issued by the Economic Planning Unit of Malaysia (UPE:40/200/19 SJ.363) and the Indonesian Institute of Sciences (LIPI: No. 2411/FRP/SM/X/2008 and No. 0115/FRP/SM/VI/2009).

Results

The MiSeq run generated a total of 1,086,926 paired-end reads after combining the data from two different libraries (prepared identically) from the same extract. Most of the fragments on the original library were 75 bp, whereas the average length of the mapped reads was 53 bp. Raw sequences were merged if possible and duplicates collapsed, producing a total of 538,995 reads, which were BLAST searched (Altschul et al., 1990) against the NCBI database. 115,266 reads were successfully assigned, and of these, 47,400 hits corresponded to bacteria, 40,751 were assigned to mammals (of which 29,720 were specifically assigned to human), 334 were assigned to Reptilia, and 48 were specifically assigned to Anolis carolinensis. Only three reads were assigned to Draco, each of which involved the mitochondrial ND2 sequence posted on GenBank for Draco cristatellus sample TNHC 56763 (see below). The paucity of Draco hits is not surprising given that there are no Draco reference genomes to map to. The extraction control was sequenced producing a total of 104,417 PE reads. After processing (adaptor removal, merged if possible, and duplicate removal) a total of 15,480 reads were assigned by the BLAST search. Bacteria were represented by 6,805 reads, mammals by 4,711 reads (of those, 3103 were assigned to human), and reptiles were assigned no reads (one read was assigned to chicken, Gallus gallus).

Of the three holotype ND2 reads, two were broadly overlapping, and the joint ND2 data obtained from the holotype totaled only 125 bp. To search for additional mitochondrial contigs in the holotype MiSeq data, we first generated a partial mitochondrial genome for TNHC 56763 from exome-capture off-target by-catch, which returned 8,114 bp of protein-coding gene sequence data. Mapping the holotype data to the TNHC 56763 mitochondrial assembly resulted in the recovery of an additional 17 reads totaling 777 bp of mitochondrial sequence data representing six genes (COI, COXIII, ATPase8, ND4, ND4L, ND5). Thus, a total of 13 contigs were assembled from 20 reads with an average length of 61 bp. Most of the mapped regions had 1 × coverage and parts of four contigs had 2 × coverage. No reads were recovered when mapping the extraction blank against the same reference. In comparing the holotype sequence data with TNHC 56763 across the 721 bp of homologous sequence data, we found that the two samples were only weakly divergent from one another, sharing the same base calls at 710 of 721 positions for a raw sequence divergence of 1.5%. We then mapped the raw reads from the exome captures for the remaining 13 D. fimbriatus group samples to the 10 D. cristatellus holotype contigs, which returned as few as two and as many as six homologous sequences per sample.

Comparison of the mitochondrial data obtained from the Draco cristatellus holotype with homologous data obtained for D. fimbriatus group samples found that the sample TNHC 56763 from Santubong, Sarawak, Malaysian Borneo was much more similar to the holotype than were any other D. fimbriatus group samples. When limiting our comparison to the six gene fragments for which we had between six and 13 corresponding sequences for comparison to the holotype, we found that TNHC 56763 was 0.9% divergent from the holotype, whereas all other samples ranged between 11.9% and 15.1% divergent. TNHC 56763 differed from the holotype at just five of 547 base positions, whereas the other samples differed from the holotype at from 36 of 303 bp to 70 of 465 base positions. The ND2 comparisons were most comprehensive because we had access to complete ND2 sequences for 65 D. fimbriatus group samples. Whereas TNHC 56763 differed from the holotype at three of 183 base positions, all other samples differed by at least 24 base positions. Notably, the two D. punctatus samples from Sarawak (the type locality for D. cristatellus) differed from the holotype at 28 and 30 of 183 base positions (15.3% and 16.4%, respectively).

Figure 1 Phylogenetic tree for the Draco fimbriatus group including the D. cristatellus holotype.

Phylogenetic tree for the Draco fimbriatus group based on a parsimony analysis of the complete mitochondrial ND2 gene (1,032 bp). The D. cristatellus holotype includes 183 bp of sequence data. Only two of 28 available D. maculatus samples were included to simplify the image. Non-parametric bootstrap values (1,000 replicates) are superimposed on the single most parsimonious phylogram for select clades. The photo in the bottom left is Draco punctatus. Photo: Jimmy A. McGuire.

A parsimony phylogenetic analysis of the ND2 gene including the D. cristatellus holotype strongly supports the monophyly of the holotype together with TNHC 56763 to the exclusion of all other D. fimbriatus group samples with 100% bootstrap support (Fig. 1). Further, the 1.6% ND2 sequence divergence between TNHC 56763 and the holotype is only slightly greater than that between samples of D. punctatus from a single locality (Sitahuis, Sumatra; uncorrected ND2 divergence of 1.2%). This does not consider the possibility that one or more of the five documented base substitutions could be sequencing errors resulting either from damage to the holotype DNA or random errors in our low coverage data, as we could not apply informatics pipelines developed to identify post-mortem damage of ancient DNA to our data (e.g., Mateiu & Rannala, 2008; Molak & Ho, 2011). Notably, one of the three inferred ND2 substitutions is a first position C->T change that would result in a proline to serine amino acid replacement, suggesting that this might be the result of post-mortem deamination of the template molecule or sequencing error.

Discussion

In the present study, we applied HTS to a 145-year old fluid-preserved holotype specimen in an effort to disentangle an otherwise intractable taxonomic question. The problem stems from the fact that one of the species in the Draco fimbriatus group, Draco cristatellus, was described using limited color information, and because fluid-preserved specimens representing multiple sympatric D. fimbriatus group species are often indistinguishable from one another without color information. Indeed, sympatric species in this complex are effectively cryptic once they have been prepared as museum specimens. This combination of circumstances rendered it virtually impossible to resolve the species status of D. cristatellus relative to D. punctatus and D. fimbriatus, two widespread species on the Greater Sunda Shelf. Importantly, a sample (TNHC 56763) collected in 1996 by JAM provides phylogenetic evidence for a third D. fimbriatus group species on Borneo, with the natural question being whether this sample is conspecific with the name-bearing holotype specimen of D. cristatellus housed in the British Museum of Natural History. Several species composition outcomes were possible, all of which were considered in a hypothesis-testing framework. In Hypothesis 1, D. cristatellus and D. punctatus are synonyms, together representing a single species distinct from D. fimbriatus and TNHC 56763. In Hypothesis 2, D. cristatellus and D. fimbriatus are synonyms. In Hypothesis 3, D. cristatellus is a species distinct from D. punctatus and D. fimbriatus but conspecific with TNHC 56763. Finally, in Hypothesis 4, D. cristatellus, D. fimbriatus, D. punctatus, and TNHC 56763 all represent distinct species, with TNHC 56763 representing a fourth sympatric species on Borneo. The only way to conclusively test these alternative hypotheses was to obtain informative genetic data from the holotype specimen of D. cristatellus for comparison with TNHC 56763, and representative specimens of D. fimbriatus and D. punctatus.

We initially believed that genetic data would easily be retrieved from the Draco cristatellus holotype. The holotype was prepared before the advent of formalin-fixation, and we consequently had reason to believe that the specimen was originally fixed with ethanol and had never been exposed to formalin. Because tissue samples collected for genetic analysis are routinely stored in ethanol, we were confident that the holotype would still hold high molecular-weight DNA suitable for genomic sequencing. Our hope was to perform exome-capture with this sample and include it in a larger Draco phylogenomic data set. However, our initial attempts at extracting DNA from the sample using methods appropriate for historical and formalin-fixed tissues failed, forcing us to adjust both our approach and our expectations. Fortunately, our alternative hypotheses proved testable without comprehensive genomic data from the holotype. Indeed, analysis of the initial 125 bp of mitochondrial ND2 data identified when the holotype sequence data was subjected to GenBank BLAST allowed us to reject hypotheses 1, 2, and 4 in favor of Hypothesis 3. The additional 596 bp of mitochondrial data obtained via mapping of holotype contigs to the reconstructed mitochondrial genome for TNHC 56763 simply provided additional confirmation that D. cristatellus and D. punctatus are each valid species, and that our specimen TNHC 56763 from Santubong, Sarawak is indeed a true D. cristatellus exemplar. This finding was a best-case scenario because all future specimens for which tissue samples are obtained can now be compared with known D. cristatellus, D. punctatus, and D. fimbriatus samples for genetic identification.

What lessons can be learned from our attempt to obtain genetic data from the Draco cristatellus holotype? First, even when initial attempts at extraction and quantification of DNA suggest that none is present, small numbers of DNA molecules may survive in the sample. For questions of simple species identification involving old and highly degraded samples, it may only be necessary to obtain limited data—even a few hundred base pairs of mitochondrial data may be sufficient to address the question. Our study shows that this is indeed possible even when initial assessments suggest that DNA in a tissue sample has been highly degraded. Obtaining data in these instances will likely require highly specialized extraction procedures such as the silica-column based extraction methodology utilized here, followed by short-fragment sequencing. Finally, we believe that the difficulty we confronted with this ethanol-fixed sample—which is consistent with the problems experienced by Ruane & Austin (2016) with their presumed alcohol-fixed samples (four of five of which failed to sequence)—suggests that older ethanol-fixed specimens might be more problematic than formalin-fixed specimens for genomic sequencing efforts (see Handt et al., 1994 for a description of DNA hydrolysis). We note that tissue samples are routinely stored in 95% ethanol for future DNA sequencing, and that standard dogma is that old museum samples fixed in ethanol should be readily amenable to DNA sequencing, whereas formalin-fixed samples are expected to be particularly challenging for sequencing efforts (see Zimmermann et al., 2008). Thus, it would be both surprising and ironic if old museum specimens originally fixed and stored in ethanol prove to be less favorable for HTS than formalin-fixed specimens. This possibility has important implications for curatorial practices. Not only is potential hydrolytic damage cause for concern with whole fluid specimens stored in ethanol, but it could also be problematic for tissue samples stored in ethanol that are not maintained in sub-zero degree conditions. Evaporation of ethanol from tissue vials might render even modern tissue samples problematical for genetic analysis.

Conclusions

The development of HTS has revolutionized biological research by making genome-scale data readily available at a reasonable cost, even for non-model organisms. Systematists have fully embraced these advances in data acquisition for freshly sampled specimens, but are just beginning to harness HTS for the millions of fluid-preserved historical samples housed in natural history collections around the world. As we have shown here, acquiring genetic data from old museum specimens will sometimes present special challenges, but the information that can be gleaned from such specimens may be the only way to conclusively resolve previously intractable evolutionary and taxonomic questions.

Supplemental Information

Supplemental Information 1 ND2 sequence data for members of the Draco fimbriatus group plus 13 mitochondrial sequence reads obtained from the D. cristatellus holotype

This file includes ND2 sequence data obtained via traditional Sanger sequencing for 39 individuals representing Draco cristatellus, D. fimbriatus, D. hennigi, D. punctatus, and D. maculatus, plus 183 base pairs of ND2 obtained via MiSeq HTS for the Draco cristatellus holotype (BMNH 1872.2.19.4). In addition, the 13 mitochondrial sequence reads obtained for the D. cristatellus holotype are provided.

Click here for additional data file.

The authors would like to thank Patrick Campbell and Donney Nicholson of The Natural History Museum, London for assistance processing the loan of liver tissue from the D. cristatellus holotype. Colin McCarthy and Oliver Crimmen kindly provided suggestions for locating information on A.H. Everett’s Bornean collections and the history of fluid preservation at BMNH, respectively. We acknowledge Alison Devault and Jacob Enk at MYcroarray and MYreads both for advice and for generating the library preparation confirming the low yield of our initial DNA extraction. David Wake, Rauri Bowie, and members of the McGuire and Bowie labs provided valuable suggestions that improved the manuscript. JAM thanks the Economic Planning Unit of Malaysia and the Indonesian Institute of Sciences (LIPI) for providing scientific collecting permits.

Additional Information and Declarations

Competing Interests

Author Contributions

Animal Ethics

Field Study Permissions

DNA Deposition

Data Availability

The authors declare there are no competing interests.

Jimmy A. McGuire conceived and designed the experiments, performed the experiments, analyzed the data, contributed reagents/materials/analysis tools, prepared figures and/or tables, authored or reviewed drafts of the paper, approved the final draft.

Darko D. Cotoras and Brendan O’Connell conceived and designed the experiments, performed the experiments, analyzed the data, authored or reviewed drafts of the paper, approved the final draft.

Shobi Z.S. Lawalata conceived and designed the experiments, analyzed the data, contributed reagents/materials/analysis tools, authored or reviewed drafts of the paper, approved the final draft.

Cynthia Y. Wang-Claypool, Xiaoting Huang, Guinevere O.U. Wogan, Sarah M. Hykin, Sean B. Reilly, Lydia L. Smith and Heather Milne performed the experiments, authored or reviewed drafts of the paper, approved the final draft.

Alexander Stubbs analyzed the data, authored or reviewed drafts of the paper, approved the final draft.

Ke Bi analyzed the data, contributed reagents/materials/analysis tools, authored or reviewed drafts of the paper, approved the final draft.

Awal Riyanto conceived and designed the experiments, contributed reagents/materials/analysis tools, authored or reviewed drafts of the paper, approved the final draft, provided key samples needed for the study.

Evy Arida, Jeffrey W. Streicher and Djoko T. Iskandar conceived and designed the experiments, contributed reagents/materials/analysis tools, authored or reviewed drafts of the paper, approved the final draft.

The following information was supplied relating to ethical approvals (i.e., approving body and any reference numbers):

The University of California, Berkeley provided full approval of all Draco-related research undertaken as part of this project.

The following information was supplied relating to field study approvals (i.e., approving body and any reference numbers):

Specimens collected in support of this research were obtained under research permits issued by the Economic Planning Unit (EPU) of Malaysia and the Indonesian Institute of Sciences (LIPI).

The following information was supplied regarding the deposition of DNA sequences:

The sequence data is available at GenBank (SRP135680) and in the Supplemental Information.

The following information was supplied regarding data availability:

The sequence data is available at GenBank (MH070349–MH070383; SRP135680) and in the Supplemental Information.

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
