# Peer review of "Squeezing water from a stone: high-throughput sequencing from a 145-year old holotype resolves (barely) a cryptic species problem in flying lizards"

_PeerJ, doi:10.7717/peerj.4470_

## Round 0.1 · original submission · Major Revisions

From what I can tell, the main issue the reviewers have is transparency and lack of detail on methods. Particularly given that this is mostly a methods paper, both reviewers feel that much more detail is needed on the methods, and I agree. One reviewer also notes much referencing of unpublished data, which I discourage.

# Staff Note: The PeerJ policy for unpublished data is that authors should avoid it if possible (by providing the data), but otherwise cite it in text
as (e.g.) "J Smith, unpublished data, 2017" #

Reviewer 1 ·

Basic reporting

The basic reporting in this article is generally very good, but some improvements could be made to the data sharing and transparency. Details can be found below.

Experimental design

The experimental design presented in this article is generally good, any shortcomings are primarily due to the nature of the historical sample.

Validity of the findings

The validity of the findings presented in this article is generally good, but some extra findings could be reported to support the authenticity of the data. Details can be found below.

Additional comments

Dear authors,

Thank you for your submission on Draco taxonomy, I think you make an important point in showing that the sequencing of holotypes can resolve taxonomic and nomenclatural issues. I have to commend you for your generally well presented publication. Please find below the comments I have relating to your submission.

Please make sure that you have accession numbers for the deposited sequence data on re-submission, these data can be uploaded to genbank with a hypothetical release date, and genbank staff will release the data earlier if appropriate.

I don’t think this paper qualifies as a “major revision”, but I would like to verify some of the changes before publication hence the recommendation as such (the requested BLAST and ancient DNA stats may or may not support the authenticity of the recovered sequences).

Yours sincerely.
* * *
Some more “major’ issues:

On many occasions the paper refers to unpublished data/findings or data/findings that will be published in the future, this does not improve transparency or the reproducibility of the research, from what I can find PeerJ does appear to approve this type of referencing though.
Line 104. “subsequent paper”, the reader can not verify your taxonomic findings and just has to assume these findings are true. But, as the section mentions this is not related too much to the objective of this paper, so leave as is.
Line 188. “unpublished data”. Again the reader just has to assume this method has been tested and can not verify its validity; the reader can not form an opinion on whether it was the extraction method or the sample that caused the lack of sufficient DNA. At least give more detail if you refer to this unpublished data/experiment.
Line 203. “unpublished data”. Exactly what part did Cotoras et al. contribute?
Line 232. “customized”. Exactly what part of SeqPrep was customised?
Line 245. “in prep”. This seems fair enough, but again please realise that the reader can not verify the validity of methods and results of this experiment because it hasn’t been described in detail.
Line 292. “unpublished data”. Please describe briefly in which way both methods contribute.
Line 304. “unpublished data”. Because Cotoras et al is not described in detail; does it actually (pr likely) improve extraction? The reader can’t verify this.
Could you please improve this overall lack of transparency?

Concerning you BLAST results I would like to see some more details.
Line 312. Here you mention 2 libraries from the same extract, but this is not mentioned in the methods section, could you please clarify? In what way do these libraries differ?
If there is only the smallest difference between these libraries (except for index adapters), could you please provide the the BLAST results per library?
Also, could you please report the BLAST results exhaustively? e.g. could you please list all organisms that received a hit? If lizards get significantly more hits than other random vertebrates this is an indicator that the sequences you have recovered are genuine and this finding should be reported. If the exhaustive list gets too big please include as supplemental material.

To further support the authenticity of the recovered ancient DNA sequences please provide some appropriate statistics. What is the average length of the recovered D. cristatellus sequences and how does this compare to the rest of the library (i.e. the contaminants). Could you give some detail as to how many and which mutations you see at the ends of the sequence reads compared to 56763? Please reference an ancient DNA paper on cytosine deamination to support your findings.

Table 1. Please remove the Total and % columns. These genes do not have equal evolutionary rates, these statistics are therefore biased due to the recovery of different regions per sample. Instead you could list the percentage for each independent measurement, e.g. “12/79 (15.2%)”. For this reason you also can’t refer to the percentages/total numbers mentioned in the paragraph starting line 339, instead you would have to mention the values per gene, this paragraph requires rewriting considering this information. Please also correct the value mention in line 356 so that you compare ND2 vs ND2 values rather than “mitochondrial genes” vs ND2 values; please also mention the values for D. punctatus.



Some minor issues:

Line 60-65. This is perhaps a matter of opinion but since your paper’s main focus is not the effect of ethanol preservation I would avoid ending your abstract on this subject. I would suggest you finish on the big picture finding i.e. your taxonomic findings and the fact that sequencing holotype specimens is important in resolving these issues.

Line 85. It is not clear what “10 of 21 specimens” refers to in this instance.

Line 158. Please do not use the word “sensible” as it expresses an opinion.

Line 216. I don’t understand how you get to 50ml, please clarify. I see 0.5 ml, 13ml, 30ml, 25ul, 25ul. Which does not add up to 50ml, is the rest lysate?

Line 277. Please rewrite this sentence, you are correct to not use ML or Bayesian approaches, but the mentioned branch lengths are not informative because your alignment is gapped, e.g. the D. cristatellus sequences appear closer to each other because of limited sequence data and therefore fewer distinguishing substitutions.

Line 281. Please also make the HTS data available. I understand that you may prefer to hold on to the capture data for release with its respective paper, so you may want to release just those sequences that mapped to the mitochondrial genome. All data from the historical specimen could be deposited in the Sequence Read Archive.

Line 282. Is it really necessary to upload this distance matrix elsewhere? I think it may be more convenient for readers if it was included as online supplement.

Line 377. Please use initials rather than “senior author”.

Line 412. “denatured”, is "degraded" perhaps more appropriate?

Line 421. I am sceptical about this conclusion. Please find some more papers that compare DNA preservation in formalin and ethanol and amend/reference this statement accordingly.

Figure 1. Although this may be obvious please describe in the figure legend which animal is depicted in the lower left picture.

Some other remarks:
I suggest using BWA instead of Geneious in the future as the algorithms employed in the latter are not sufficiently described.

·

Basic reporting

Language and structure are appropriate overall (though note exceptions detailed below). Figures are appropriate. Raw data is supplied, though I'd also like to see the HTS data available on Dryad.

Experimental design

Research is original and the research question is well defined. However key details and justification of the methods are not provided by the authors, which mean that I cannot recommend the manuscript be accepted/published in its present state (see details below).

Validity of the findings

I have no major issues with the conclusions of this study. I think the data, though admittedly scarce, are ultimately adequate to address the research question.

Additional comments

Summary:

The authors use a variety of approaches to obtain mtDNA from the holotype specimen on D. cristatellus, a flying lizard, in order to verify its taxonomic validity. The authors successfully obtain limited data from the holotype, and find that it is genetically distinct from closely related congeners while being closely related to a recently collected specimen also identified as D. cristatellus. Thus, the validity of D. cristatellus is supported. The authors also comment on the possible implications of the methodological difficulties they encountered for future studies using non-ideally stored specimens, and indeed much of the paper is focused on this aspect of the work.

I don't have any any major criticisms of the results of this manuscript or their interpretation (insofar as sufficient detail has been given to allow me to assess them). However, given the methodological focus/pitch of the paper there is a distressing lack of detail and critical examination of many of the steps undertaken, particularly with regard to the HTS. As my background is in HTS, and particularly HTS as applied to ancient/degraded samples, this is the area of the paper that I am best equipped to assess and many of my comments/criticisms reflect this. In my view, there is too little documentation and too many unexplained observations (e.g. extremely high human contamination in sample and extraction control) for me to recommend the acceptance of this manuscript in its current form.

Main issues:

1. Overall lack of details in the method section, particularly regarding the HTS and PCR on the D. cristatellus holotype.

2. Splitting of information between Methods and Results that makes comprehension extremely difficult, as the reader is forced to flick back and forth between the two to actually figure out what the authors have done. All of the information pertinent to the Methods should be moved from the Results section to the Methods section.

3. Unexplained and distressing human contamination in the HTS data, and lack of acknowledgement of the ancient DNA laboratory involved...

Detailed comments:

Line 104 - "For reasons that we will describe in a subsequent paper"
Though it may not ultimately be critical for interpreting the results of this paper, I would prefer that the authors give some preliminary justification for this decision. What assurance is there that this subsequent paper will actually be published, or that the reader of this will be able to find said paper? Something like "Based on X (Unpublished Data and/or Pers. Comm.) we instead..." might be more appropriate. The following sentence (beginning Line 106) regarding the type locality of D. fimbriatus does nothing to clarify this issue for me, though I think that is the intent.

Line 106 onwards - I'm confused about whether the D. fimbriatus mentioned is D. fimbriatus sensu McGuire et al. or D. fimbriatus sensu Manthey. This issue persists in the final paragraph of the introduction where I'm unclear on whose taxonomy is being used. The primary aim of the paper seems to be verifying the distinctness of D. cristatellus, which does not appear to be affected by this taxonomic issue. So why do the authors not leave unraveling the D. fimbriatus/punctatus/abbreviatus mix-up for their subsequent paper where it can be dealt with fully and explicitly, and thereby save the reader of this paper any undue confusion? I do understand that the authors are trying to "future-proof" this manuscript, but as written currently I think it just muddies the water for someone unfamiliar with the detailed taxonomy of the group (i.e. me!).

Line 190 - What ND2 primers, PCR recipe, and cycling conditions were used for this amplification? I'm surprised that the authors were able to amplify human DNA using primers designed for reptile ND2 (it would be less odd for the conserved 12S/16S genes) and I'm not aware of any universal primers for ND2. If the authors were not using taxon-specific primers (or indeed, if they were targeting too long a fragment) then it is not surprising that the amplification failed. Perhaps the annealing temperature was too low? Insufficient detail is given here for a reader to assess the reasons for failure. Similarly, in this paragraph the authors mention "low" DNA yield but don't quantify this or even state what method was used to quantify the DNA (though I note this this is later reported in the Results section).

Line 198-199 - Which ancient DNA lab exactly? As far as I can see they aren't even listed in the acknowledgements... Did the lab specifically ask not to be named?

Line 231-232 - What parameters were used in SeqPrep?

Line 233-234 - The authors note they assessed the quality of sequences using FastQC, but we are left to guess whether quality was good or bad because the authors make no statement about this. I assume that the quality was OK...

Line 255-257 - How many unique reads? What proportion of the reference was covered? What was the average read depth? What was the average read length?

Line 264-267 - How were the contigs for the D. cristatellus holotype generated? Why map all of the other sequencing reads against those contigs instead of mapping everything against the Draco reference you built using the data for TNHC56763 (including the D. cristatellus data)?

Line 280 - I strongly suggest that the authors also make their HTS data available on the Dryad repository.

Line 290-309 - This whole paragraph is basically a repetition of information in the Methods, so I think this paragraph should be removed and any non-redundant information moved to the relevant section of the Methods.

Line 313-316 - I have to believe there is some mistake here. The authors have 538,995 reads of which 29,790 are assigned to human. That means 5% of the DNA molecules in the DNA extract are human! Orders of magnitude more than the actual target lizard DNA. Is it possible that there was some mix-up of barcodes with a human DNA library run in parallel? Because otherwise this represents an extremely distressing level of contamination in an ancient DNA laboratory. It could perhaps be written off as contamination of the sample (maybe through handling by museum staff) except that ~3% of molecules in the extraction control also appear to be human! I would appreciate a statement from the authors on this issue, and should also hope that the aDNA lab is currently investigating this further and reviewing their protocols as this level of contamination could easily compromise any mammal research (not just human) being undertaken in the lab.

Line 325 - Could the authors please report mean read length and depth for the D. cristatellus holotype data?

Line 362-364 - This transition almost certainly represents post-mortem deamination of the template molecule (as ubiquitously observed in ancient/museum/degraded samples). Given the low number of reads sequenced in the present study, any transitions observed in the D. cristatellus holotype should be viewed with suspicion.

Line 376-389 - Does this refer to published work? Or is this another reference to the authors' upcoming publication? Either way, this information would have been more appropriate in the Introduction, with a brief summary at the beginning of the Discussion. Why are specific hypotheses only being outlined in the Discussion?

---

## Round 0.2 · Minor Revisions

I feel both reviewers have again provided valuable comments on the paper and I side with reviewer 1 in the recommendation to modify the columns of the table. I also think the comments on the last paragraph of the paper as well as the details of human contamination and details of blast hits are important and should be addressed if at all possible.

Reviewer 1 ·

Basic reporting

I would like to thank the authors for their improvements to the manuscript, particularly the addressing of unpublished data/methods has much improved the paper.

I believe I may not have been clear enough concerning the reasons behind exhaustively reporting the BLAST hits. I completely agree that the mammal, human and bacterial contaminants do not pose any significant problems and are commonplace (DNA from domesticated/human associated animals is often encountered in these experiments). However, showing (by means of a BLAST list) that you receive significantly more hits on Iguana (Anolis+Draco) compared to other totally random animal (sub-)orders (e.g. Lamniformes) would actually add further support that your data are genuine and not some artefact. I leave it up to you if you want to incorporate this idea.

With respect to the holotype data; I would like to see the average total library length vs the average mapped length for both the capture and the shotgun sequencing experiments. These data are a good indicator whether or not your data are genuine, normally the overall libraries are expected to be longer due to recent/human contaminants in comparison to the genuine DNA that was fragmented.

With regard to removing the "total" and "%" columns in the table. I respectfully disagree that readers will be aware of the effect of evolutionary rates on gene divergence, part of your audience (think morphological systematics and field biologists) may not fully appreciate the biases in these numbers (also note that pairwise nucleotide distances in ND2 are about 75% more than in COI simply due to differing rates). I therefore strongly suggest that you include a line along the lines of "the divergence estimates are affected by varying evolutionary rates and are therefore indicative. the divergence estimates do however strongly indicate [your conclusion]".

With regard to ethanol preservation both your paper as well as that of Ruanae and Austin make an intersting observation that deserves mentioning. Note that Ruanae and Austin only looked at a very limited 4-5 alcohol preserved specimens, with an average age older than the formalin fixed specimens. Altogether there is not a lot of evidence for superior formalin DNA preservation over that of ethanol preservation. There are far more papers that suggest alcohol preservation outperformes formalin in terms of DNA preservation though (https://doi.org/10.1016/S0002-9440(10)64472-0 https://dx.doi.org/10.1186%2F1742-9994-5-18 https://doi.org/10.3897/zookeys.196.3130 just to pick a few), this therefore deserves some balanced reporting backed up by references.

Please also correct the value mentioned in line 344 so that you compare ND2 vs ND2 values rather than “mitochondrial genes” vs ND2 values; please also mention the values for D. punctatus.

Experimental design

please see the basic reporting section

Validity of the findings

please see the basic reporting section

·

Basic reporting

No additional comments.

Experimental design

No additional comments.

Validity of the findings

No additional comments.

Additional comments

The authors have satisfactorily addressed most of the questions and concerns I raised in my original review.

I have only three remaining comments, only one of which (the third) really requires any action or response from the authors. Otherwise, I think this manuscript is suitable for publication in PeerJ in its current form.

1)
The last paragraph of the introduction remains extremely dense, with binomials and authors - that the casual author will be completely unfamiliar with - flying thick and fast. At the end of the introduction I am still left unclear as to the exact taxonomic hypotheses being tested. The authors give an excellent and clear description of all competing hypotheses, however it does not appear until the beginning of the discussion. If it were me, I would simply move the first paragraph of the discussion to the end of the introduction. This straightforward change requires no additional rewriting, but would allow the reader to "catch up" if they became lost in the dense taxonomic discussion at the end of the introduction and gives them clear context for the Results section. I understand that this comes down to a difference of opinion and style, so I'll leave the resolution of this issue to the editor's discretion.

2)
The authors partially addressed my concern about human contamination in the DNA extractions. Indeed it's quite plausible that the contamination in the sample itself comes from handling by museum staff/researchers. What remains unclear to me is why the extraction blank control should exhibit similarly high levels of contamination. Regardless, I agree with the authors that it is highly unlikely to have impacted the results of their study, as no lizard contamination was detected. It's simply a curiosity that I hope the ancient DNA lab in question is monitoring or following up on independently, ~3% human DNA in a blank control seems a lot.

3)
Line 288-299 - "after combining the data from 2 different libraries (prepared identically) from the same extract"

In the Methods section (a) (Lines 229-230) the authors describe only one library being created from the holotype sample (and one from the extraction blank control). However, in the line quoted above they refer to two different libraries from the same extract. Could the authors please make clear exactly how many libraries were created and sequenced from each sample/control, and make sure this is consistently reported throughout?

---

## Round 0.3 · accepted · Accept

You did a nice job answering the reviewers comments.